Journal of Data-centric Machine Learning Research (2024)          Submitted 3/24; Revised 5/24; Published 6/24

# NAFlora-1M: Continental-Scale High-Resolution Fine-Grained Plant Classification Dataset

**John Park**                                          JPNONMOON@GMAIL.COM
*The New York Botanical Garden, Bronx, NY, USA*
*SCINet Program and ARS AI Center of Excellence, Office of National Programs, USDA Agricultural*
*Research Service, Beltsville, MD, 20705, USA*

**Riccardo de Lutio**                        RICCARDO.DELUTIO@GEOD.BAUG.ETHZ.CH
*EcoVision Lab, Photogrammetry and Remote Sensing*
*ETH Zurich, 8092 Zürich, Switzerland*

**Brendan Rappazzo**                                      BHR54@CORNELL.EDU
*Cornell University*
*Ithaca, NY, USA*

**Barbara Ambrose**                                      BAMBROSE@NYBG.ORG
*Laboratory*
*The New York Botanical Garden, Bronx, NY, USA*

**Fabian Michelangeli**                                  FABIAN@NYBG.ORG
*Institute of Systematic Botany*
*The New York Botanical Garden, Bronx, NY, USA*

**Kimberly Watson**                                      KWATSON@NYBG.ORG
*Herbarium*
*The New York Botanical Garden, Bronx, NY, USA*

**Serge Belongie**                                      S.BELONGIE@DI.KU.DK
*Pioneer Centre for Artificial Intelligence*
*University of Copenhagen, 1350 Copenhagen, Denmark*

**Damon P. Little**                                      DLITTLE@NYBG.ORG
*Lewis B. and Dorothy Cullman Program for Molecular Systematics*
*The New York Botanical Garden, Bronx, NY, USA*

Reviewed on OpenReview: *https://openreview.net/forum?id=UIOM1SSJd0*

**Editor:** Sergio Escalera

## Abstract

The plant kingdom exhibits remarkable diversity that must be maintained for global ecosystem sustainability. However, plant life is currently disproportionately disappearing at a rapid rate, putting many essential functions—such as ecosystem production, resistance, and resilience—at risk. Plant specimen identification—the first step of plant biodiversity research—is heavily bottlenecked by a shortage of qualified experts. The botanical community has imaged large volumes of annotated physical herbarium specimens, which present a huge potential for building artificial intelligence systems that can assist researchers. In this paper, we present a novel large–scale, fine–grained dataset, NAFlora-1M, which

consists of 1,050,182 hebarium images covering 15,501 North American vascular plant species (90% of the known species). Addressing gaps from previous research efforts, NAFlora-1M is the first–ever dataset to closely replicate the real–world task of herbarium specimen identification, as the dataset is intended to cover as many of the taxa in North America as possible. We highlight some key characteristics of NAFlora-1M from a machine learning dataset perspective: high–quality labels rigorously peer–reviewed by experts; hierarchical class structure; long–tailed and imbalanced class distribution; high image resolution; and extensive image quality control for consistent scale and color. In addition, we present several baseline models, along with benchmarking results from a Kaggle competition: A total of 134 teams benchmarked the dataset in a total of 1,663 submissions; the leading team achieved an 87.66% macro-$F_1$ score with a 1–billion–parameter ensemble model—leaving substantial room for future improvement in both performance and efficiency. We believe that NAFlora-1M is an excellent starting point to encourage the development of botanical AI applications, thereby facilitating enhanced monitoring of plant diversity and conservation efforts. The dataset and training scripts are available at `https://github.com/dpl10/NAFlora-1M`.

**Keywords:** Biodiversity, Plant Diversity, Plant Specimen Collection, Plant Specimen Images, Digitization, Herbarium, Fine-grained Image Classification, High-resolution Images, Long-tail Distribution, Class Imbalance, Class Hierarchy, Hierarchical Label, Annotation Quality, Image Quality Control, Kaggle Competition

## 1 Introduction

The plant kingdom exhibits remarkable diversity, with a staggering 434,934 documented extant land plant species (Pimm and Joppa, 2015; Enquist et al., 2019). Maintaining plant diversity is essential for global ecosystem sustainability, such as ecosystem productivity (Tilman et al., 2012), resistance (Isbell et al., 2015), and resilience (Oliver et al., 2015). The rapid decline in global biodiversity (Butchart et al., 2010; Urban, 2015) poses an extreme threat to plant community conservation, since many plant taxa are rare and therefore difficult to find (Enquist et al., 2019). A recent UN report (IPBES, 2019) highlights a concerning trend in biodiversity loss: over a million species are now at risk of extinction, with plant life disproportionately disappearing at a rapid rate.

Historically, over 3,000 herbaria around the world serve as the hub for physical repositories of more than 396 million specimens (Thiers, 2022) collected throughout human history (Davis, 2023). These plant specimens—standardized collections of pressed, dried, and mounted plants—serve as valuable reference in plant diversity research, especially for plant species identification (Thiers et al., 2016; Soltis, 2017; Thiers, 2020).

Plant specimen identification—the applied science of determining the proper names of plants—is regarded as the first and foremost step of all biodiversity studies. As Linnaeus once said, "If the names are unknown, knowledge of the things also perishes" (Stafleu, 1971). Correct determination of name of organisms is prerequisite of all subsequent studies; e.g., modeling species occurrence in space and time; land management and invasive species monitoring; discovering new species. All of these applications require correct and timely specimen identification—a pressing need in light of the rapid global biodiversity decline (Butchart et al., 2010; Urban, 2015; IPBES, 2019).

However, the manual examination of these herbarium specimens is extremely time–consuming due to a shortage of qualified experts. For instance, the average time between

collection and identification spans several months to a few years. Discovery of new species takes even more time—on average, 35 years (Bebber et al., 2010).

In recent decades, the use of digital technology to mobilize physical herbarium specimens online has emerged as a significant advancement in the field of biodiversity science (Hedrick et al., 2020). The botanical community has so far "digitized" (imaged physical herbarium specimens) and made publicly available a total of 45.3 million specimens of 85,800 plant taxa (GBIF: `https://www.gbif.org/`; `https://doi.org/10.15468/dl.yh72t2`). Although the digitization of these herbarium specimens has promoted unprecedented research collaborations (Davis, 2023), the speed of specimen identification—whether physical or digital—is still quite slow using the current manual system that heavily relies on a limited number of trained experts.

One potential way to address this challenge is to develop an automated system, exploiting the massive volume of digitized collections. Deep learning–based image classification is one of the most robustly studied fields in artificial intelligence, therefore application to real–world problems, such as plant diversity, could be a showcase example. However, training deep neural network classifiers for a new domain may require massive amounts of annotated images—which poses a great challenge. Producing a massive annotated image dataset often requires trading annotation label quality for volume, which has been an issue for standard datasets such as ImageNet (Shankar et al., 2020; Tsipras et al., 2020).

Digitized hebarium specimens are excellent data sources to address this challenge, since the sheer volume of expert–annotated data is available as the byproduct of century–long plant diversity research. collectively, the world's herbaria possess millions of specimen images, of which label annotations—usually the most labor–intensive and time–consuming part for deep learning data collection process—are publicly available. Moreover, these are high quality label annotations, as the identification of collected plants are cross–checked by multiple experts for their distinctness and consistency over a long period of time (e.g., centuries; Bebber et al., 2010; Thiers et al., 2016; Soltis, 2017; Hedrick et al., 2020; Thiers, 2020; Davis, 2023). Leveraging the large number of expert–verified annotated images, development of accurate automated specimen classifier can alleviate the bottleneck of manual identification, thus allowing us to address the rapid global biodiversity decline in a more timely manner.

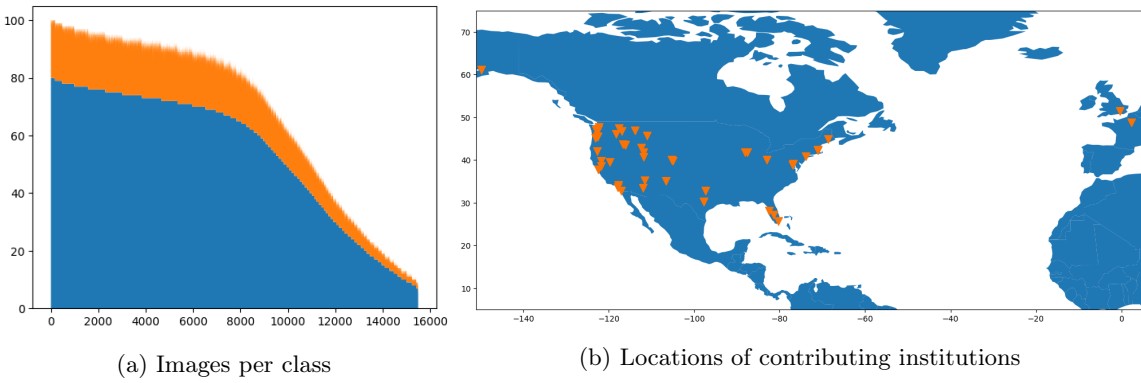

(a) Images per class          (b) Locations of contributing institutions

Figure 1: **The frequency distribution of the NAFlora-1M dataset.** (a) The number of images (y-axis) for training (blue) and testing (orange) by species (x-axis). (b) Institutions contributing to the dataset.

Recently, a handful of studies (Carranza-Rojas et al., 2017; Shirai et al., 2022; Little et al., 2020; de Lutio et al., 2021) have attempted to build deep plant specimen classifiers, by assembling a subset of image data from the large pool of available herbarium images worldwide. Initial studies were limited to few model architectures and training methodologies (Carranza-Rojas et al., 2017; Shirai et al., 2022). Little et al. (2020) used a deep learning competition to explore a wide–range of state–of–the art deep learning architectures and training techniques for herbarium specimen images, but the dataset was too small. de Lutio et al. (2021) explored a unprecedented number of taxa (64.5k) and images (2.50M), however their dataset resulted in low species coverage (14%) for the sampled geographical scope.

We argue that for training plant specimen classifiers, it is crucial to create a dataset that closely matches the real–world use cases of plant diversity research. By and large, considering geographic and taxonomic context is critical to a great degree in plant diversity, since all life forms have evolved to adapt to their own environment—consequently, each geographic area has a distinct collection which accounts for all plant life, called a "flora". In order to better serve the intended user–audience (i.e., botanists), a dataset which covers finite geographic and taxonomic range as explicitly and completely as possible is of utmost importance. Neural networks trained on such datasets can directly contribute to on–going research projects, which are normally bound to particular geographic regions. In contrast, a model trained on datasets without regard to geography or taxonomy is less applicable, as predictions for out of distribution input data are not reliable.

To this end, we built a geographically all–inclusive herbarium specimen image classification dataset. Our dataset, NAFlora-1M, is a large–scale, fine–grained dataset, which explicitly and completely covers a finite geographic region: North America. NAFlora-1M closely resembles the real–world application of plant species identification, as it attempts to encompass the flora of a well–known distinct geographic region.

Important NAFlora-1M characteristics include (specifics in the following section 2):

- High–quality labels: annotations have been repeatedly peer–reviewed by domain experts; each class is organized into multiple hierarchical levels (e.g., family, genus); classes are mutually exclusive; and each image nominally contains objects from only one class.

- High–resolution images: the long axis of each image is 1,000 pixels.

- Standardized images: scale, color, and background are well–controlled.

- Long–tail distribution: naturally collected plants present imbalanced classes.

Another important aspect we considered for developing NAFlora-1M was size. As enumerated above, herbarium specimen images have significantly different characteristics, compared to other general–purpose image classification datasets (e.g., ImageNet). Optimizing neural network architectures and training require an adequate size of benchmarking dataset specific to the research domain. We attempt to achieve adequate size of the dataset whilst maximizing plant representations for North America. The dataset was sampled from a larger pool of 8,776,687 images available from 176 institutions covering a total of 17,241 plant species (methods in section 2). Our final dataset consists of 1.05M herbarium specimen images representing 15.5k distinct species, encompassing more than 90% of North American vascular plants (Figures 1 and 2). The dataset size resembles that of mid–size datasets,

such as ImageNet, which has been served as a reference dataset for benchmarking visual recognition tasks in computer vision, since 2011 (Deng et al., 2009).

Extensive data cleaning, high–quality labels, and high–resolution images are rare for large–scale fine–grained classification datasets, but without these characteristics, models may have difficulty differentiating between noise and subtle inter–class features. NAFlora-1M is a unique large–scale fine–grained classification dataset that will benefit both computer vision, by allowing development of novel methodologies, and botany by providing a practical tool. We present broad–scale benchmarking of NAFlora-1M on modern vision architectures conducted via the Herbarium 2022 Kaggle competition. The top competition submissions were analyzed in order to identify factors that improve model performance.

## 2 The NAFlora-1M dataset

**Label integrity**  Plant specimen collections in herbaria can be several hundred years old, hence are invaluable documentation of plant diversity, physical structure, and geographic and temporal occurrence (Thiers, 2020). The collections consist of pressed, dried, and mounted specimens with additional specimen metadata detailing morphology, phenology, distribution, genomic information, etc. (Figure 2). Professional botanists curate the specimens

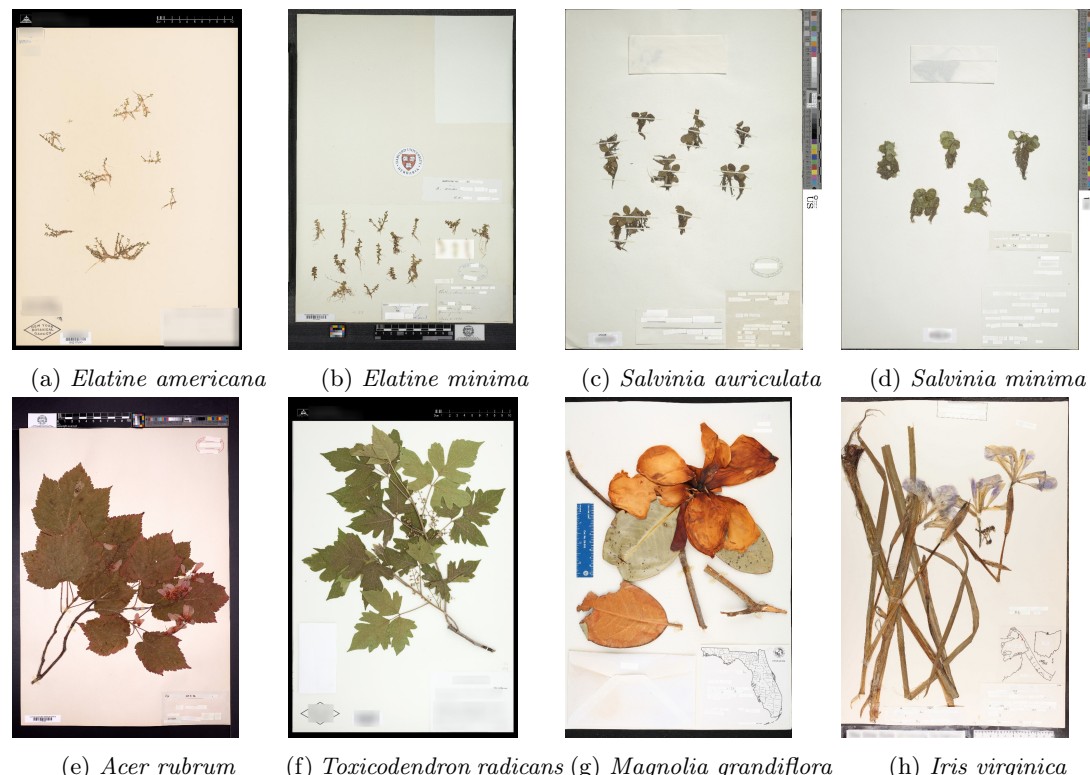

(a) *Elatine americana*  (b) *Elatine minima*  (c) *Salvinia auriculata*  (d) *Salvinia minima*

(e) *Acer rubrum*  (f) *Toxicodendron radicans*  (g) *Magnolia grandiflora*  (h) *Iris virginica*

Figure 2: **Subtle and obvious variation in plant features found in NAFlora-1M**. (a–d): minute plants that require fine–grained analysis due to their small overall size; (e–h): larger plants that are more straightforward to identify at low–resolution.

continuously to maintain their archival integrity and update the specimen metadata (Thiers, 2020). Plant species, genera, etc. are peer–reviewed for their distinctness and consistency by multiple experts, ensuring the ability to create high–quality fine–grained datasets. NAFlora-1M possesses the unique characteristic that each image nominally contains one class label, which is advantageous for learning class–specific features.

**Label hierarchy**  The plant names (labels) used by botanists are arranged in a hierarchy (i.e. a family contains one or more genera, a genus contains one or more species). This hierarchy has been designed such that it mirrors the reconstructed evolutionary events (i.e. divergence, extinction, etc.) that produced the plant diversity currently observed. Each hierarchic level has an unique (reconstructed) origin [i.e. families and genera are monophyletic *sensu* Farris (1974)] thereby maximizing the information content and predictivity of each name (Farris, 1979).

**Image characteristics**  Digital herbarium specimen images are created with a resolution of up to 9,000×6,000 pixels. The resolution varies depending on the institution, device, and project (Soltis, 2017), with an average resolution of around 3,700×3,000 pixels. NAFlora-1M images were resized such that longest edge is 1,000 pixels—which is still much higher than other fine–grained vision datasets (Figure 2). The standardized image–capture protocols used ensure consistency in color, background, and scale—this uniformity is ideal for representation learning. The majority of computer vision datasets contain light contamination and background noise which make it difficult for models to learn fine–grained differences. While herbarium specimen images can be normalized for consistent scale (the vast majority of images include a physical scale bar) many other computer vision datasets consist of images with a variable, and unknown, scale. Extreme variability in scale can affect neural network transfer learning performance (Van Horn et al., 2021).

**Scalability**  Herbarium specimen image datasets have the potential to be scaled without additional labeling cost: worldwide herbaria currently curate 400M physical specimens that already have peer–reviewed annotations—providing a wealth of potential data for assembling novel fine–grained image datasets. 24M plant specimens representing 93k species have already been digitized and made publicly available (typically CC0, CC BY, or CC BY-NC licenses). Annotation quality often trades-off with dataset size in web-scraped machine learning datasets (e.g., ImageNet annotation quality has challenged number of times (Tsipras et al., 2020; Shankar et al., 2020), and improving the annotation quality demonstrated better outcome (Ridnik et al., 2021)). In addition, plant specimens are mostly free from societal issues associated with crowd–sourced collection pipelines such as selection biases, offensive or hateful annotations, and privacy violations (Peng et al., 2021; Yang et al., 2020).

**Dataset construction**  In field expeditions for plant diversity research, plants are collected, pressed, and dried in the field. Once they arrive at research facilities they are identified by researchers—usually specialists in a particular flora or group of plants. After species identification, each specimen is mounted on blank archival herbarium paper. Generally specimens are databased and digitized before insertion into the collection. The herbarium specimen images obtained from each source institution are created following strict, standardized protocols (Thiers et al., 2016).

The first step of the data collection for NAFlora1M involved downloading these "digitized" specimen records and images. We obtained a complete updated list of North American (Canada, Greenland, and United States of America) vascular plants from the "Checklist of the Vascular Plants of the Americas" (CVPA; Ulloa Ulloa et al., 2017). Then we retrieved records for the 17,041 vascular plant species from the two largest public biodiversity aggregators— GBIF (Global Biodiversity Information Facility; Telenius, 2011) and iDigBio (Soltis, 2017)— after standardizing the names via the "World Checklist of Vascular Plants" (WCVP; v5; Govaerts et al., 2021). In total, the matching records resulted in 8,776,687 images belonging to the 17,041 vascular plant species. Among them, plant species with a low number of available images were removed ($n \leq 10$) and the maximum number of images for the species retained was capped at 100 (for species with more than 100 images available, 100 images are randomly selected for download). Additionally, to restrict the amount of custom download code required, we selected records to maximize the number of species with at least 10 images while at the same time minimizing the number of download servers. Consequently, we select a total of fifteen download servers which provide images for 54 herbaria.

Finally, as a post–processing step, text information was removed from all images, to make sure the only mode being utilize is vision, and to nullify any privacy concerns. We employ the Character Region Awareness for Text (CRAFT) detection method (Baek et al., 2019) to detect image regions that contain text. We applied uniform noise and a Gaussian blur filter to the text–detected regions. This process ensured that any text with metadata was obfuscated from the model—forcing the model to focus solely on specimen differences in terms of visual perception. After text information removal, we resized the images to 1,000 pixels in the largest dimension. In the end, NAFlora-1M has 1,050,179 images from 54 institutions (Figure 1b) representing 90.06% of the species known from North America (15,501 species; Figure 1a). The dataset is partitioned with 80/20% split, resulting in 839,772 training images and 210,407 testing images, under the rule to include at least two images for each class in the test partition. More specifics about dataset construction are detailed in the supplementary material.

**Baseline experiments and ablation studies** The purpose of baseline experiments is to provide fair comparisons among the existing neural architectures. We chose a total of five different baseline backbones, based on reputation, size, efficiency, and architecture type—the standard ResNet-50 (He et al., 2016) for a reference model, based on its popularity; MobileNetV3 (Howard et al., 2019) for representing compact neural networks; Efficient-NetV2 (Tan and Le, 2021) to represent a training cost–efficient architecture. Among the different variants of the family, EfficietNetV2-S is the most comparable to ResNet-50 in terms of model size and training throughput. In addition, we also included ViT-B/32 and ViT-B/16 to represent transformer architecture. All networks were pre–trained on ImageNet-1k and fine–tuned with $256^2$ pixel training images for 30 epochs. In terms of data augmentations we simply used two: random horizontal–flip and random rotation of 15 degrees. We utilize a cyclical learning rate scheduler (Smith, 2017), which has fast convergence with CNNs (Table 1); 20 epochs with a cyclical learning rate scheduler has shown to produce faster convergence than 100 epochs of a constant learning rate (Smith and Topin, 2019). To allow sufficient iterations for model convergence, models were trained for 10 more epochs (total of 30 epochs). CNNs were optimized using Stochastic Gradient Descent with Weight decay

(SGDW) and the Cross Entropy (CE) loss with label smoothing (= 0.1). ViT-B/16 and B/32 were fine–tuned with AdamW optimizer, following Touvron et al. (2021) for learning rate and weight decay settings. We applied the same loss function with the same label smoothing threshold to ViT. We conducted several ablation studies to assess the effects of network size, input size, and loss functions. An example training and inference script can be found on GitHub[1].

**Kaggle competition benchmark** Benchmarking via a Kaggle competition provided an opportunity to rigorously test NAFlora-1M with a large number of participants from all over the world. The competition was held for 15 weeks from February 14 to May 30 in 2022. It followed the ILSVRC format where the 210,407 testing images were divided into two equal sets for public validation and final evaluation. Teams provided top-1 predictions for each testing image and macro-$F_1$ scores were used to rank teams. In total, 134 teams contributed to the benchmarking the data with over 1,000 submissions (Tables 5 and 6).

**Training an efficient model** Based on results from the Kaggle competitions, we refine the training strategy of our model. We choose to base our final model on the EfficientNetV2-S backbone, due to the good performance–to–cost trade–off it displays in the baseline experiment (Table 1). We scale the the input image size up to $400^2$ pixel images for achieving better performance. The final neural network is finetuned with $400^2$ pixel images for 60 epochs on NAFlora-1M (macro-$F_1$ score = 80.47%), with cyclical learning rate scheduler (Smith and Topin, 2019), and the maximal learning rate scaling linearly with the batch size as suggested by Goyal et al. (2018). The details can be found in the Supplementary A.2 and in the GitHub repository `https://github.com/dpl10/NAFlora-1M/blob/main/src/naflora1m_train_and_infer.py`. It took about 20 hours to train the network on Google TPU v3-8. The final model is used for the post–hoc analysis.

**Post–hoc analyses** To determine which factors influence the classification performance of models on NAFlora-1M, an instrinic factor—super–class identification is examined via an ANOVA test on the genus– and family–level predictions to determine if significant interfamilial and intergeneric variation in performance is present. Second, an extrinsic factor, the number of training images ($n$), is explored, by aggregating the accuracy for all species within binned intervals based on $n$, in increments of 10 (i.e., {[5, 10), [10, 20), [20, 30), ..., [80, 90)}.

## 3 Results

### 3.1 Baseline experiments and ablation studies

Among the backbones tested, the EfficientNetV2 family present the best performance–efficiency trade–off for NAFlora-1M (Table 1). ResNet-50 depicts 70.79% macro-$F_1$ when trained on $256^2$ input images, whereas EfficientNetV2-S shows 78.36% macro-$F_1$ under comparable training cost (Table 1). MobileNetV3 and EfficientNetV2-B1 show little training cost difference (minutes/epoch: 8.08 vs. 8.50) despite the large difference in the model size (3M vs. 7M), with substantial performance gap between the two (macro-$F_1$: 57.50% vs. 75.80%), ViT-B/32, which has a similar number of FLOPs to ResNet-50 (Dosovitskiy et al.,

---

1. `https://github.com/dpl10/NAFlora-1M/blob/main/src/naflora1m_train_and_infer.py`

| Model | Params | Train | Macro-$F_1$ | Time per | ImageNet-1k | |
| Name | (M) | Pixels | | Epoch (min) | Train | Top-1 Acc. |
|---|---|---|---|---|---|---|
| MobileNetV3-S[a] | 3 | $256^2$ | 57.07 | 8.08 | $224^2$ | 68.1 |
| ResNet-50[b] | 24 | $256^2$ | 70.79 | 11.12 | $224^2$ | 75.30 |
| EfficientNetV2-S[c] | 20 | $256^2$ | **78.36** | 11.64 | $300^2$ | 83.9 |
| ViT-B/32[d] | 86 | $256^2$ | 69.96 | 11.18 | $224^2$ | 81.66[e] |
| ViT-B/16[d] | 86 | $256^2$ | 74.99 | 29.78 | $224^2$ | 83.97 |

Table 1: **Baseline performance of neural architectures on NAFlora-1M.** The top score is set in **bold face**.

a. Howard et al. (2019)

b. He et al. (2016)

c. Tan and Le (2021)

d. Dosovitskiy et al. (2020)

e. `https://github.com/google-research/vision_transformer`

| Model | Params | Train | Macro-$F_1$ | Time per | ImageNet-1k | |
| Name | (M) | Pixels | | Epoch (min) | Train | Top-1 Acc. |
|---|---|---|---|---|---|---|
| EfficientNetV2-B1[a] | 7 | $256^2$ | 75.52 | 8.50 | $240^2$ | 79.8 |
| EfficientNetV2-S[a] | 20 | $256^2$ | 78.36 | 11.64 | $300^2$ | 83.9 |
| EfficientNetV2-M[a] | 53 | $256^2$ | **79.97** | 20.73 | $380^2$ | 85.1 |

Table 2: **Ablation study of model size.** The top score is set in **bold face**.

a. Tan and Le (2021)

2020), performs and costs similar as well (Table 1). The slightly lower performance of ViT-B/32 compared to ResNet-50 (Table 1) corresponds to reports of vision transformers requiring more training resources (Dosovitskiy et al., 2020). The limited scope of our experiment did not determine the vision transformer performance ceiling, however, the ViT-B/16 result indicates that smaller patches results in better performance (69.96% vs. 74.99%) at the cost of greater training resources (about $3\times$ slower; Table 1). The Kaggle competition results, detailed below, hint at the vision transformer performance gains possible with increased training resources.

Ablation studies of model size and image resolution give us some idea about the performance increase of models in two different cases: increasing model size (Table 2) is beneficial for increasing model performance, but comes with greater training cost. For example, it takes twice as longer to train EfficientNetV2-M than EfficientNetV2-S (Table 2)—although the performance gain is clear (79.97% vs. 78.36%). Increasing training pixel size has a comparable effect to increasing network size (Table 3), but it turns out to be more time efficient in our experiment. Loss function has a small, but noticeable, effect with class balance loss producing better performance (Table 4).

| Train Pixels | Macro-$F_1$ | Time per Epoch |
|:---:|:---:|:---:|
| $256^2$ | 78.36 | 11.64 |
| $320^2$ | **79.51** | 16.60 |

Table 3: **Ablation study of input resolution.** The top score is set in **bold face**.

| Loss Functions | Macro-$F_1$ |
|:---:|:---:|
| CE loss | 78.36 |
| Class balance loss[a] | **78.71** |

Table 4: **Ablation study of loss functions.** The top score is set in **bold face**.

[a]. Cui et al. (2019)

## 3.2 Backbones used in Kaggle competitions

The Kaggle competition produced a NAFlora-1M benchmark across a diverse set of backbones (at least 12; Table 5), image resolutions ($224$–$672^2$ pixels), loss functions, augmentation methods, and post–processing methods. Over the course of 15 weeks, a total of 134 teams produced 1,663 submissions. The top-5 teams achieved macro-$F_1$ scores from 85.14% to 87.66% with 301–1,300M parameter ensemble models. These experiments required substantial multi–GPU training hours—an invaluable investment in NAFlora-1M. A summary of the top submissions is presented in Table 6.

One of the most remarkable characteristics of the top teams was sheer model scale—the majority employed ensembles totaling more than 1-billion parameters (Table 6). It is also notable that Vision Transformers (ViT) and their variants—such as Swin-Transformer (Liu et al., 2021), DeiT-III (Touvron et al., 2022), Cswin (Dong et al., 2022), Swin-Transformer-V2 (Liu et al., 2022a), and Meta-Transformers (Diao et al., 2022) are the dominant types of models in this competition. Whereas in previous competitions, including the Herbarium 2021 Challenge (de Lutio et al., 2021), CNNs, such as SE-ResNeXt (Hu et al., 2018) and EfficientNet (Tan and Le, 2019), mainly served as general backbones. This shows the high performance of transformers as general backbones for image classification tasks and demonstrates that subsequent transformer variants, such as Swin-Transformers (Liu et al., 2022a, 2021), have overcome shortcomings since revealed in the original ViT (Dosovitskiy et al., 2020). Notably, some CNNs persisted in the competition—such as RegNets (re–parameterized SE-ResNeXt; Radosavovic et al., 2020), ResNeSt (Zhang et al., 2022), and ConvNeXt (Liu et al., 2022b). ResNet variants, such as RegNetY (Radosavovic et al., 2020) and ResNeSt (Zhang et al., 2022), showed stable performance as they were the top-4 and the top-5 teams' main backbones (Table 6). ConvNeXt (Liu et al., 2022b) appears to be a powerful backbone—comparable to Swin-Transformer (Liu et al., 2021). In summary, the best performing single models in the Herbarium 2022 competition were (descending macro-$F_1$ score): SwinV2-B, Swin-B, Swin-L, Metaformer, ConvNext-B, scwin-L, RegNetY-12G, DeiT-III-B, EffNet-B6, ResNeSt-101. This is in contrast to the previous competition where the list

| Model | Params (M) | Pre–train Input Size | IN-1k[a] Top-1 Acc. | NAFlora-1M Input Size | NAFlora-1M Macro-$F_1$ | Pre–train dataset |
|---|---|---|---|---|---|---|
| CSwin-L[b] | 173 | $384^2$ | **87.5** | $384^2$ | 85.01 | IN-22k[c] |
| Swin-L[d] | 197 | $384^2$ | 87.3 | $384^2$ | 85.61 | IN-22k |
| SwinV2-B[e] | 88 | $384^2$ | 87.1 | $384^2$ | **86.28** | IN-22k |
| ConvNeXt-B[f] | 89 | $384^2$ | 86.8 | $384^2$ | 85.96 | IN-22k |
| DeiT-III-B[g] | 87 | $384^2$ | 86.7 | $384^2$ | 84.95 | IN-22k |
| Swin-B[h] | 88 | $384^2$ | 86.4 | $384^2$ | 86.05 | IN-22k |
| EfficentNet-B6[i] | 43 | $528^2$ | 86.1 | $512^2$ | 84.32 | Noisy[j] |
| ResNeSt-101[k] | 48 | $256^2$ | 83.0 | $672^2$ | 84.90 | IN-1k |
| RegNetY-12G[l] | 84 | $224^2$ | 80.3 | $544 \times 416$ | 84.80 | IN-1k |

Table 5: **Model backbone information reported for the Kaggle competition.** Top scores are set in **bold face**.

a. ImageNet-1k
b. Dong et al. (2022)
c. ImageNet-22k
d. Liu et al. (2021)
e. Liu et al. (2022a)
f. Liu et al. (2022b)
g. Touvron et al. (2019)
h. Liu et al. (2021)
i. Tan and Le (2019)
j. Noisy: pre–trained model with noisy student method described in Xie et al. (2020).
k. Zhang et al. (2022)
l. Radosavovic et al. (2020)

was composed of ResNeSt-101, ResNeXt-101, TResNet, GENet, NFNet, SE-ResNeXt-101, SE-ResNet50 (de Lutio et al., 2021).

### 3.3 Training cost

Training many large backbones and aggregating them into ensembles may produce the most robust and reliable performance; however, the computational cost is huge. Thus, it is unlikely to be the most computationally efficient solution, as the performance gain per unit training cost is marginal. It required over 1,000 multi–GPU hours for some team's training effort (Table 6)—if these efforts were measured in cloud computing dollars, they could easily exceed $50k. Much of this large computational overhead probably does not come from learning representations from images as a large portion of model parameters are in the classification head: one–hot encoding of the 15k species alone requires ∼30M parameters. Even with compact feature extractor, such inflated classification head will inevitably cause a large parameter model—which can be even more pronounced in an ensemble. In a competition setting, even small improvements in performance can make a big difference in leader board ran, however, from the perspective of an end–user, such as a botanist, it may not be practical to follow the recipe of the winning team due to the difficulty of getting access to the high–cost multi–GPU clusters. Therefore, research focused on plant specimen image classification

| Rank | Team 1 | Team 2 | Team 3 | Team 4 | Team 5 |
|---|---|---|---|---|---|
| Macro-$F_1$ score | 87.66% | 87.09% | 87.03% | 85.25% | 85.14% |
| Genus-level macro-$F_1$ | 96.37% | 96.47% | 96.14% | 95.26% | 95.04% |
| Family-level macro-$F_1$ | 97.92% | 98.12% | 97.98% | 97.21% | 96.89% |
| Backbones used | ResNeSt-101 EffNet-B6 DeiT-III-B Swin-B, -L ConvNeXt-B CSwin-L SwinV2-B | Metaformer-0 to 2 ConvNext-T, B, L | Swin-B, -L CSwin-B, -L BeiT-B, -L | ResNeSt-101 Swin-L | RegNetY-12G |
| Ensemble | 8 | 12 | 6 | 2 | 4 |
| Train pixels | $512^2$, $384^2$ | $384^2$, $448^2$ | — | $672^2$, $448^2$ | $224^2$, $448^2$, $512^2$, $544{\times}416$ |
| Parameters | 940M | 1,300M | 1,040M | 302M | 338M |
| Loss functions | Species: sub–center Arcface & CE. Genus & family: CE | Seesaw loss on species and genus labels | — | CE with label smoothing = 0.1 | Adaptive Mining Sample and CE |
| Augment-ation methods | Same as Liu et al. (2021) | AutoAugment (Cubuk et al., 2018) Color jitter Random erasing | Same as Liu et al. (2021) | Same as Liu et al. (2021) | AutoAugment (Cubuk et al., 2018) Random flip Random resized crop |
| Post processing | Ensemble aggregation | Five crop inverse frequency weighted prediction | Ensemble aggregation | Random resized crop $\times$ 100 summed up top-5 | Ensemble aggregation Five crop |
| Total Estimated Training Time | — | 1200 hours | — | — | 200 hours |
| Hardware Used | 4$\times$A-100 | 8$\times$A-100 | 8$\times$A-100 | 8$\times$RTX-3090 | V-100 |

Table 6: **Model architectures and training techniques for the top-5 competition submissions.**

should specifically include efforts to improve the performance–to–cost trade–off in order to promote increased use of these models within the botanical community.

### 3.4 Addressing class imbalance

Several teams tested new loss functions to address the data imbalance (Table 6), for instance, the top–ranked team incorporated the sub–center Arcface loss (Deng et al., 2020)—a method developed to tolerate label errors by establishing multiple sub–categories within each class. The predecessor of sub–center Arcface loss, Arcface loss(Deng et al., 2019), is very effective at differentiating large numbers of classes and has been shown to be even more effective than range loss (Zhang et al., 2016)—which was designed to address long–tail distribution problem. The second–ranked team implemented Seesaw loss (Wang et al., 2021), which was designed to address class imbalance, by dynamically rebalancing the gradients of positive and negative samples for each category (Wang et al., 2021). The fifth–ranked team incorporated Adaptive Mining Sample loss (AMS) which was designed for person re–identification (Huang et al., 2022). In addition, our ablation study utilized class balanced loss (Cui et al., 2019) to address long–tail distribution of the data.

There are other methods to address data imbalance, for example, over–sampling or under–sampling techniques—however, both present significant risks. For oversampling, repeated samples could cause model overfitting (Cui et al., 2019). An alternative way to increase minor class samples is by interpolation (Chawla et al., 2002) or synthesis (Zou et al., 2018), but it is unclear if these synthetic data are independently and identically distributed samples from the

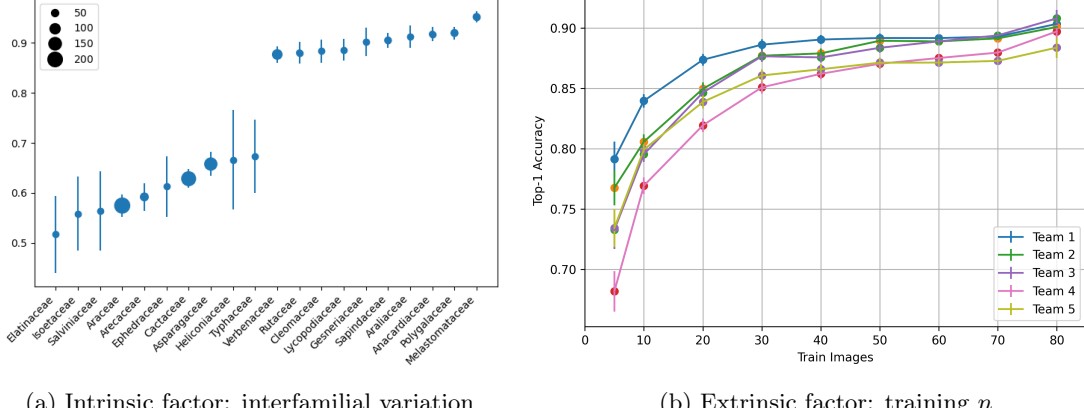

(a) Intrinsic factor: interfamilial variation    (b) Extrinsic factor: training $n$

Figure 3: **Intrinsic and extrinsic factors that influence NAFlora-1M performance.**
(a) The circle sizes represent number of test images in each family; error bars indicate
standard errors. We show the Top-10 and the bottom-10 families here in terms of average
Top-1 accuracy. (b) The error bars indicate standard errors of average Top-1 accuracy,
aggregated over the test images within each intervals.

true population. In the end, synthetic datasets are drawn from training data distribution and
the noise applied to provide independence could easily be transmitted to error–proneness (Cui
et al., 2019). Under–sampling is preferred (Drummond et al., 2003), however it has the risk
of removing valuable samples for learning representations within the data (Cui et al., 2019).
According to Cui et al. (2019), both under–sampling and over–sampling techniques have
downsides, thus loss functions designed specifically to tackle class imbalance appears to be
best practice.

### 3.5  Class hierarchy

A number of teams attempted to take advantage of the label hierarchy by combining the
losses from different taxonomic levels: the top–ranked team incorporated fully connected
layers for all three hierarchic levels while the second–ranked team used classification layers
for species and genus labels. The hierarchical labels supplied in NAFlora-1M are a synthesis
of our current understanding of plant evolutionary history—based on numerous analyses of
phenotypic and genomic sequence data (Govaerts et al., 2021). The labels are not designed
to reflect raw visual similarity, but there is a strong correlation between overall appearance
and evolutionary history (Farris, 1979). Although computer vision models do not necessarily
perceive images in the same manner as botanists, Table 6 provides indirect evidence that
they are correlated: models exclusively trained on species labels (teams 3–5) have the same
rank order for species–level, genus–level, and family–level macro-$F_1$ performance. In addition,
the performance of these models improves at higher hierarchical levels—just like that of
human botanists.

### 3.6 Input image size

Modern architectures and training strategies allow for low–resolution training—such as $128^2/160^2$ pixels—that is able to obtain results as good, or better, than their predecessors (Bello et al., 2021; Touvron et al., 2019). In contrast, the top-5 teams (macro-$F_1$ score: 85.15–87.66%) employed high image–resolutions ranging from $384^2$ to $672^2$ pixels. Many modern model architectures and scaling strategies are optimized for ImageNet performance using techniques such as Neural Architecture Search (NAS)—an automatic layer selection and scaling strategy for input resolution, layer depth, and layer width. Due to the lack of public large–scale datasets, rigorous studies of model scaling for datasets with distinct characteristics, such as high–resolution images or classes that require fine–grained analysis are conspicuously absent from the literature.

### 3.7 Factors affecting model performance

It is notable that classification performance varies significantly among families (Figure 3a). From a botanist's perspective, some plants are intrinsically difficult to differentiate due to factors such as fine–grained characteristics, small size, and the need to observe particular, often fleeting, life phases (e.g., flowers). Even for specialists, it is challenging to identify minute species, such as genus *Elatine* (Figures 2a and 2b) in the family Elatineceae and genus *Salvinia* (Figures 2c and 2d) in the family *Salviniaceae*. Elatinaceae (average accuracy 51.69%) and Salviniaceae (average accuracy 56.43%) are two of the least accurately classified plant families among the 272 plant families in NAFlora-1M (Figure 3a). For some species, the lack of characteristic leaf variation combined with the absence of other identifying parts can greatly complicate identification: for instance, *Iris virginica* (Figure 2h) is quite easy to identify using distinct floral characteristics, but if flowers are absent, species–level identification is nearly impossible from leaf characteristics alone. On the other hand, *Acer* (Figure 2e) in the family Sapindaceae and *Toxicodendron* (Figure 2f) in the family Anacardiaceae have leaf characteristics that make them easier to identify without flowers, which explains the high average accuracies in Sapindaceae (90.53%) and Anacardiaceae (91.76%) (Figure 3a).

The number of training images, $n$, is arguably the most difficult extrinsic factor to control: it has a significant impact on accuracy—particularly when $n$ is small (Figure 3b). Once $n$ reaches ∼40, the effect begins to diminish and accuracy begins to saturate. The greatest difference (10.95%) in mean top-1 accuracy between the best– and worst–performing team is for plant species with $n \in [5, 10)$, whereas the difference is minimal (2.84%) for species with $n \in [40, 50)$. Mitigating the loss in accuracy for species with small $n$ is key to higher competition ranking. The difference in accuracy between $n \in [80, 90)$ and $n \in [5, 10)$ is similar to team rankings (Team 1: 11.2%; Team: 13.32%; Team 3: 17.51%; Team 4: 21.53%; Team 5: 14.95%). Intriguingly, the fifth–ranked team performed better than the third– and fourth–ranked teams for the $n \in [5,10)$ interval—perhaps due to the ability of AMS loss (Huang et al., 2022) to cope with imbalance classes (Table 6). The accuracy of the fifth–ranked team (87.27%) was lower than that of the other teams in the interval containing the majority of species ($n \in [70, 80)$; 6,352 species).

## 4 Conclusion

Artificial intelligence is advancing rapidly, and there is a growing interest in testing algorithms against domain experts to assess the feasibility of automating routine processes. Given the urgent biodiversity crisis, exasperated by climate change, biology is one of the most crucial use cases. A significant reduction in global biodiversity is predicted (Urban, 2015), making it increasingly necessary to identify and monitor a wide range of species at increased spacial and temporal resolutions. Thanks to the digitization of natural history collections and the aggregation of those digital data, everyone has direct digital access to specimens collected worldwide (Nelson and Ellis, 2019). The National Science Foundation's ADBC program has provided over $100 million in funding for digitization projects at more than 300 institutions in the United States since 2010 (National Academies of Sciences et al., 2020). NAFlora-1M is built upon this effort and will enable vision researchers to test classification methods on high–quality, fine–grained data whose labels are have been peer–reviewed by plant systematists and at the same time, botanists will be able to identify plants more rapidly—enhancing biodiversity monitoring and potentially helping to predict the trajectory of future changes.

NAFlora-1M falls into fine-grained visual recognition dataset in computer vision. It can provide computer vision researchers in evaluating the performance of their model architectures and training techniques. The introduction of vision transformers demonstrates that large models with reduced inductive bias—via the long–range dependent attention mechanism—benefit significantly from massive quantities of training data (e.g., JPT-300M; Dosovitskiy et al., 2020). The current prevailing benchmark method focuses on ImageNet-1k top-1 accuracy—making it impossible to determine how well model architectures and training techniques will perform on real–world fine–grained tasks. NAFlora-1M provides an alternative benchmark—particularly well–curated for fine–grained classification tasks.

On the other hand, we have highlighted a number of obstacles to training a full–accuracy model on NAFlora-1M. The inefficiency of the best–performing models in terms of model size and compute cost is a major challenge. In the end, the winning teams come from large tech companies that provide ample access to computer power. In resource limited conditions, it is challenging for biologists—who have a great need for models of this type—to follow the top–ranked teams' training protocols. Consequently, it is essential to determine the best balance between model performance and compute cost. This work provides a starting point for training a relatively light–weight model with manageable computing costs (20 hours on TPU v3-8) that is able to deliver decent performance (i.e., 80.47% macro-$F_1$ score).

## 5 Acknowledgements

Cloud TPU access from Google's TPU Research Cloud (TRC) and funding from the National Science Foundation (USA; DEB 2054684) is gratefully acknowledged.

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
