# OpenReview forum: "NAFlora-1M: Continental-Scale High-Resolution Fine-Grained Plant Classification Dataset"
_DMLR — Accepted by DMLR_

### Review · Reviewer_b4U1 · 2024-03-30

**Recommendation:** 3
**Confidence:** 1

**Summary Of Contributions:**

The paper presents NAFlora-1M, a new dataset for continental-scale, high-resolution, fine-grained plant classification, featuring over one million images of 15,501 North American vascular plant species, which accounts for 90% of known species in the region. This dataset aims to closely replicate the real-world task of herbarium specimen identification, with images mostly sourced from U.S. herbaria.

**Strengths:**

1. The paper proposes a novel large-scale, fine-grained dataset, NAFlora-1M.
2. The paper presents baseline models, along with benchmarking results from a Kaggle competition.
3. This paper is well organized and clearly written.

**Audience:**

Yes

**Claims And Evidence:**

NA

**Datasets And Benchmarks:**

Yes. Authors are encouraged to make the dataset open-access.

**Extended Submissions:**

NA

**Limitations:**

1. The format of tables should be adjusted. They are out of the main pages.
2. The details of baselines should be introduced. The reason for choosing them should be analyzed.
3. More recent ViT backbone should be included in the performance comparison.

**Requested Changes:**

See above.

---

### Review · Reviewer_ZFFy · 2024-04-08

**Recommendation:** 3
**Confidence:** 2

**Summary Of Contributions:**

"NAFlora-1M: Continental-Scale High-Resolution Fine-Grained Plant Classification Dataset" introduces NAFlora-1M, a large-scale dataset for herbarium specimen identification. It consists of over 1 million images covering 15,501 North American vascular plant species. Key features include high-quality labels peer-reviewed by experts, hierarchical class structure, long-tailed and imbalanced class distribution, high image resolution, and extensive image quality control. The dataset is benchmarked via a Kaggle competition with participation from 134 teams, highlighting areas for future improvement in performance and efficiency.

**Strengths:**

S1 Comprehensive Dataset:

This covers a vast range of North American plant species, which is beneficial for biodiversity studies.


S2 Multiple Formats:

The dataset includes high-resolution images with controlled scale and color, enhancing machine learning applications.


S3 Extensive Benchmarking:

Detailed benchmarking via Kaggle competition provides a clear performance baseline for future models.

**Audience:**

Yes

**Claims And Evidence:**

The claims made in the submission are supported by convincing and clear evidence.

**Datasets And Benchmarks:**

I failed to find some related Repos

**Extended Submissions:**

No

**Limitations:**

W1 Limited Geographical Scope:

NAFlora-1M is region-specific, focusing solely on North American flora. This geographical specificity might limit the dataset's utility in global biodiversity and plant identification studies.

W2 Class Imbalance and Long-Tailed Distribution:

The dataset exhibits a long-tailed distribution with imbalanced classes. While this reflects real-world scenarios, it poses challenges for machine learning models, potentially leading to biased performances towards more common species.

W3 Limited Analysis of Hierarchical Labels:

The dataset includes hierarchical labels (species, genus, family), but the paper lacks a thorough exploration of how these hierarchical structures are utilized in model training and their impact on performance.

**Requested Changes:**

Comparison with Similar Datasets:

Broaden the discussion to include a comparative analysis with other relevant datasets in plant classification, such as "PlantCLEF," "PlantVillage," and "Extended Agriculture-Vision." This comparison could provide context on how NAFlora-1M stands out in terms of size, diversity, resolution, and challenges.


Detailed Analysis of Class Imbalance:

Expand the discussion on strategies to address the class imbalance and long-tailed distribution in NAFlora-1M. Suggesting methods like synthetic data generation, resampling techniques, or advanced loss functions would add value.

---

### Review · Reviewer_MiVK · 2024-04-16

**Recommendation:** 4
**Confidence:** 2

**Summary Of Contributions:**

This work introduces a new dataset, NAFlora-1m, for herbarium specimen identification. The dataset contains over 1 million images of 15k plant species in North America, from 54 different herbaria. Labels were peer-reviewed by experts with hierarchical class structure, and images are high resolution with quality control related to scale and color. In addition, the work includes baseline models and a summary of results from a Kaggle competition that involved 134 teams. The dataset and training scripts are made available online.

**Strengths:**

* The work is well-motivated with useful contextualization of the role of herbaria and the utility of the proposed dataset
* A Large diversity of institutions contributing to the dataset.
* Useful metadata is included in the dataset, including institution and evolutionary distance.
* The paper includes thorough details about the data collection process, image pre-processing, and experimental set-up, which will help in reproducibility.
* The work contains very detailed analysis of top submissions of the Kaggle challenge, including discussion of different models, loss functions, image size, and post processing. In addition, detailed analysis of strengths of different kinds of approaches.
* The work includes analyses of certain families of class focusing on variation in size or fine-grained details, or prevalence of time-dependent elements that are useful for distinguishing between classes.
* The authors identify of future areas of research focused on performance and efficiency improvements of classification models
* The paper includes extensive details in supplementary work about data collection and discussion of anticipated noise in the dataset.

**Audience:**

Yes

**Claims And Evidence:**

Yes

**Datasets And Benchmarks:**

Yes, although it would be useful to include discussion of whether there is any plan for adding in new species as they are identified, or updating the dataset if the ground truth taxonomization changes

**Extended Submissions:**

N/A

**Limitations:**

* The current work contains useful discussion of controls among different data sources. It could be strengthened with additional discussion of the amount of variation in herbaria source within a given class. In other words, is it likely that all images for a given class come from just one or two herbaria? In that case, there may be risks of spurious correlations with different kinds of backgrounds, lightings, cropping, or annotation schemes shown in the image.
* There could be additional details about class hierarchy, which may be useful especially for non-experts in the space.
* It would be useful to include discussion of whether there is any plan for adding in new species as they are identified, or updating the dataset if the ground truth taxonomization changes

**Requested Changes:**

* [minor] typo in line 151: "dlataset size in web-scarped"

---

### Review · Reviewer_emCS · 2024-04-19

**Recommendation:** 3
**Confidence:** 2

**Summary Of Contributions:**

This paper introduces a new large-scale fine-grained dataset for plant classification, together with expert-reviewed labels and high-resolution images. This work also brings about experiments from the Kaggle competition with a thorough analysis. This paper is of interest to specific computer vision applications and the botany community.

**Strengths:**

- The general motivation is sound and clear, where the identification of herbarium specimens is challenging even for well-trained experts. The concept of flora makes sense to me and this work covers most parts of North America and from different contributing institutions, which signifies the contribution and impact.

- The quality of the labels seems to be high-level and peer-reviewed by experts. Also the images are high-resolution, which makes the fine-grained classification feasible and tractable for complex models.

- The distribution of images is well normalized and with long-tail characteristics, which makes the research community easy to follow yet challenging enough.

**Audience:**

Yes

**Claims And Evidence:**

Yes

**Datasets And Benchmarks:**

Yes

**Extended Submissions:**

N/A

**Limitations:**

- In the experiments, although Kaggle competition provides extensive models and results, it is not clear how individual factor affects the performance, e.g. input resolution, backbones, loss functions. It seems to be more like a report for Kaggle competition if the raw results are directly analyzed. More specific ablation studies are needed.

- In Figure 3(b), it is not clear why all the models get better performance with 70-80 training images because it seems that the data complexity has been saturated from 30-70.

- The Results section looks messy to me. It needs more subsections to clearly illustrate the analysis and discussion.

**Requested Changes:**

See Limitations.